# The 4-α-Glucanotransferase AcbQ Is Involved in Acarbose Modification in *Actinoplanes* sp. SE50/110

**DOI:** 10.3390/microorganisms11040848

**Published:** 2023-03-27

**Authors:** Sophia Nölting, Camilla März, Lucas Jacob, Marcus Persicke, Susanne Schneiker-Bekel, Jörn Kalinowski

**Affiliations:** Microbial Genomics and Biotechnology, Center for Biotechnology, Bielefeld University, 33615 Bielefeld, Germany

**Keywords:** *Actinoplanes*, acarbose, acarviosyl metabolites, microbial secondary metabolite, acarbose 4-α-glucanotransferase, AcbQ, α-1,4-glucans

## Abstract

The pseudo-tetrasaccharide acarbose, produced by *Actinoplanes* sp. SE50/110, is a α-glucosidase inhibitor used for treatment of type 2 diabetes patients. In industrial production of acarbose, by-products play a relevant role that complicates the purification of the product and reduce yields. Here, we report that the acarbose 4-α-glucanotransferase AcbQ modifies acarbose and the phosphorylated version acarbose 7-phosphate. Elongated acarviosyl metabolites (α-acarviosyl-(1,4)-maltooligosaccharides) with one to four additional glucose molecules were identified performing in vitro assays with acarbose or acarbose 7-phosphate and short α-1,4-glucans (maltose, maltotriose and maltotetraose). High functional similarities to the 4-α-glucanotransferase MalQ, which is essential in the maltodextrin pathway, are revealed. However, maltotriose is a preferred donor and acarbose and acarbose 7-phosphate, respectively, serve as specific acceptors for AcbQ. This study displays the specific intracellular assembly of longer acarviosyl metabolites catalyzed by AcbQ, indicating that AcbQ is directly involved in the formation of acarbose by-products of *Actinoplanes* sp. SE50/110.

## 1. Introduction

One in ten adults live with the chronic disease diabetes mellitus. In 2021, over 536 million adults worldwide were affected, and the trend is rising [1,2]. Therefore, pharmaceutical research and drug development are of public interest and have high relevance. To fulfill the demand for treatment of all patients in the upcoming years, industrial production of diabetes drugs has to be improved [2]. The most common type worldwide is type 2 diabetes (90%), which can arise from an unhealthy lifestyle (e.g., diet, physical inactivity) and leads to inadequate insulin production [2]. Type 2 diabetes can be treated with α-glucosidase inhibitors such as acarbose, a natural product from *Actinoplanes* sp. SE50/110 [3,4]. Acarbose is commonly termed as pseudo-tetrasaccharide because it consists of an acarviosyl moiety (acarviose) and a maltose molecule connected by an α-1,4-glycosidic bond (Figure 1). The acarviosyl moiety is composed of an aminohexose bound to a C_7_-cyclitol by a N-glycosidic bond [3,5].

In the acarbose biosynthesis gene cluster in *Actinoplanes* sp. SE50/110, 16 out of the 22 genes are assigned to the acarbose biosynthesis [6]. In recent years, several investigations have been carried out to elucidate the exact biosynthesis pathway and the specific functions of the Acb proteins in the pathway [6,7,8,9,10,11,12]. In 2022, Tsunoda et al. [10] were able to experimentally verify the last part of the pathway. The C_7_-cyclitol moiety is synthesized from *sedo*-heptulose 7-phosphate by eight Acb proteins (AcbCMOLNUJR) [6,7,8,9,10,11]. In parallel, the aminohexose moiety is formed from glucose 1-phosphate by three other Acb proteins (AcbABV) and linked to maltose by AcbI [10,12]. In a final step, both units are attached to each other catalyzed by AcbS [10]. Furthermore, three Acb proteins (AcbKPQ) could not be assigned directly to the acarbose biosynthesis and are therefore more likely to be involved in modifying the product. The function of the acarbose 7-phosphotransferase AcbK has already been proven; it catalyzes the phosphorylation of acarbose [13]. The specific function of AcbP and AcbQ and their influence on acarbose synthesis is still unknown, although AcbQ as putative acarbose 4-α-glucanotransferase represents an interesting enzyme in terms of acarbose modification and by-product formation [14].

Modification and elongation of natural products is very common in many species [15]. In addition to acarbose, *Actinoplanes* sp. SE50/110 produces related acarviosyl metabolites (α-acarviosyl-(1,4)-maltooligosaccharides) with different sugar residues at the reducing end of the aminohexose (C-1 position) and/or at the non-reducing end of the C_7_-cyclitol (C-4 position) (Figure 1, Table 1) [14,16,17]. The formation of different acarviosyl metabolites is partly dependent on the carbon source, and enzymes such as AcbQ could have an important impact, resulting in a broad range of by-products [18,19,20]. AcbQ is a cytosolic protein, suggesting that modification of acarbose and acarviosyl metabolites by AcbQ takes place in the cell near the membrane [21]. Proteome analysis has shown that AcbQ features one of the highest protein stabilities during acarbose production compared to other Acb proteins. Its stability is post-translationally influenced, suggesting a high relevance in acarbose metabolism [22]. At a structural level, AcbQ has the highest similarity to MalQ from *Haemophilus influenzae*, which is a 4-α-glucanotransferase (EC 2.4.1.25) and belongs to the glycosyltransferase family [14,15]. MalQ plays an essential role in intracellular maltose and maltodextrin pathways in almost all organisms [23]. The enzyme catalyzes the hydrolysis of α-1,4-glycosidic bonds of linear glucans, releases glucose from the reducing end and transfers the glucanosyl residue to a non-reducing end of an α-1,4-glucan acceptor [24,25,26].

In this study, we analyzed the specific function of AcbQ and identified several similarities with MalQ. We tested glucans with different glycosidic bonds as substrates performing in vitro assays with AcbQ. Furthermore, the pseudo-tetrasaccharide acarbose and phosphorylated acarbose (acarbose 7-phosphate), which is mainly present in the cell [27], were tested as acceptors to analyze the formation of elongated acarviosyl metabolites. In this way, the influence of AcbQ on the formation of by-products during acarbose synthesis in *Actinoplanes* sp. SE50/110 was evaluated.

## 2. Materials and Methods

### 2.1. Strains and Cultivation Conditions

*Escherichia coli* (*E. coli*) DH5α was used for DNA manipulation and *E. coli* JM109 was used for recombinant protein production. All *E. coli* strains are listed in Appendix A and were cultivated either with LB medium or on LB agar plates including ampicillin (100 µg mL^−1^) at 37 °C.

### 2.2. Construction of the Expression Systems

The genes *acbQ* and *acbK* were codon-optimized for expression in *E. coli*, and synthesized by Invitrogen (Thermo Fisher Scientific, Waltham, MA, USA) (Appendix A). The genes were amplified by PCR with the Phusion High-Fidelity PCR Master Mix (Thermo Fisher Scientific) with corresponding oligonucleotides (Appendix A). The PCR products were subsequently purified with GeneJET Gel Extraction Kit (Thermo Fisher Scientific). The expression vector pJOE5751.1 was isolated with GeneJET Plasmid Miniprep Kit (Thermo Fisher Scientific) and digested with restriction enzymes (FastDigest BamHI and FastDigest HindIII, Thermo Fisher Scientific). The digested vector was assembled with the PCR products by Gibson Assembly (protocol from New England Biolabs, Ipswich, MA, USA) (Appendix A). The inserts of the final vectors were verified by sequencing. The expression vectors pJOE5751.1-*acbQ* and pJOE5751.1-*acbK* were transformed into *E. coli* JM109 by heat shock transformation (Appendix A). Positive transformants were selected on LB agar plates containing ampicillin.

### 2.3. Preparation of Recombinant Proteins

*E. coli* JM109 strains harboring *acbQ* and *acbK* genes were cultivated in a 10 mL pre-culture overnight. A main culture (200 mL) was inoculated with an OD_600_ of 0.1 and cultivated at 37 °C and 200 rpm. When an OD_600_ of 0.6–0.8 was reached, the temperature was reduced to 16 °C. L-Rhamnose (0.2%) was added as inducer, and the cultivation was continued for 20 h at 16 °C. The cells were harvested by centrifugation (10 min, 5500× *g*, 4 °C), the supernatant was removed, and the cell pellet was dissolved in cooled LEW buffer from the protein purification kit used (Protino Ni-TED kit, Macherey-Nagel, Düren, Germany). All following steps were performed on ice. The cells were disrupted in tubes with Zirconia beads (0.1 and 0.5 mm in size, Carl Roth, Karlsruhe, Germany) using a homogenizer (Precellys 24 homogenizer, Bertin Technologies, Montigny-le-Bretonneux, France). The disruption protocol included three steps at 6500 rpm for 30 s and cooling steps on ice for 5 min in between. To remove the cell debris, a centrifugation step was performed (20 min, 21,000× *g*, 4 °C). The recombinant proteins were purified by affinity chromatography via their His_6_-tag. AcbQ and AcbK were concentrated, and the buffer was changed with LEW buffer from the kit using Amicon Ultra centrifugal filters (10 and 30 kDa MWCO, Merck Millipore, Burlington, MA, USA). The protein purification was confirmed by discontinuous SDS-PAGE and protein concentration was determined using ROTI Nanoquant (Carl Roth) with bovine serum albumin (BSA) solution as the protein standard.

### 2.4. Preparation of Substrates and Standards

The substrates glucose (G1), maltose (G2), maltotriose (G3), sucrose (Suc) and trehalose (Tre) were purchased either from Carl Roth, Sigma Aldrich (St. Louis, MI, USA) or VWR (Radnor, PA, USA). Maltotetraose (G4), maltopentaose (G5), maltohexaose (G6) and maltoheptaose (G7) were obtained from Megazyme (Bray, Wicklow, Ireland). Acarbose (Acb) was produced and kindly provided by Bayer AG (Leverkusen, Germany). The substrate acarbose 7-phosphate (Acb-7P) was prepared enzymatically from acarbose and ATP using AcbK [13]. The reaction mixture (500 µL) containing 25 mM Tris-HCl (pH 7.5), 10 mM MgCl_2_, 20 mM NH_4_Cl, 10 mM ATP and 5 µM AcbK was incubated at 30 °C for 6 h. The enzyme was removed by ultrafiltration using an Amicon Ultra centrifugal filter (3 kDa MWCO, Merck Millipore) and the product was dried in a speed vacuum concentrator (Eppendorf Concentrator 5301, Eppendorf SE, Hamburg, Germany). Afterwards, the product was dissolved in a specific amount of water and verified by LC-ESI-MS, using the method described below (Section 2.9).

### 2.5. In Vitro Reaction of AcbQ and Analysis

In vitro reactions of AcbQ were either performed in 20 µL reaction mixtures to be analyzed by TLC and HPAEC-PAD analysis or in 100 µL reaction mixtures to measure acarviosyl metabolites by LC-ESI-MS. Each reaction mixture contained 20 mM Tris-HCl (pH 7.5), 5 mM MgCl_2_, 10 µM AcbQ and 10 mM of each substrate added to the reaction mixture and was incubated at 30 °C for 5 h. G2, G3, G4 and Acb were tested as substrates. In addition, combinations of G3 with G2, G4 and G5 and combinations of Acb with G2, G3 and G4, respectively, were analyzed. A reaction mixture with inactivated AcbQ (100 °C for 10 min) was used as a control. In vitro reactions were directly analyzed by TLC or HPAEC-PAD analysis, as described below (Section 2.6 and Section 2.8). In vitro reactions of AcbQ for LC-ESI-MS analysis (Section 2.9) were filtered using an Amicon Ultra centrifugal filter (10 kDa MWCO, Merck Millipore) to remove the enzyme.

### 2.6. Thin-Layer Chromatography Analysis

Thin-layer chromatography (TLC) was performed to separate the glucans using silica gel 60 F_254_ plates (Sigma Aldrich) as the stationary phase and a mixture of 1-butanol, 2-propanol, ethanol and water (3:2:3:2) as the mobile phase [28]. For visualization, the TLC plate was sprayed with a solution of 4% (*w*/*v*) sulfuric acid in methanol and heated the plate at 150 °C for 3 min.

### 2.7. Initial Velocity of AcbQ with Different Substrates

To analyze the substrate preference of AcbQ, the initial velocity was obtained. A reaction mixture containing final concentrations of 50 mM potassium phosphate buffer (pH 7.0), 25 mM MgCl_2_, 2 mM ATP, 2 mM NADP, 2 U hexokinase (Sigma Aldrich), 2 U glucose-6-phosphate dehydrogenase (Alfa Aesar by Thermo Fisher Scientific) and 5 mM substrate each (listed below) was preincubated at 30 °C [29]. The absorbance at 340 nm was measured by a Tecan Infinite M200 microplate reader (Tecan Group AG, Männedorf, Swiss) with the i-control 10.1 software (Tecan Group AG). When no changes of absorbance A_340_ were observed, 2 µM AcbQ was added to achieve a final assay volume of 200 µL and to initiate the enzyme reaction. The assay was performed at 30 °C for 30 min. G2, G3, G4, G5, G6, Tre, Suc, Acb and Acb-7P were tested as substrates. In addition, combinations of Acb with G2, G3 G4, Suc and Tre, and Acb-7P with G2, G3 and G4, were analyzed. All substrate combinations were measured in triplicate. A calibration curve of glucose standards was prepared to calculate concentrations.

### 2.8. HPAEC-PAD Analysis

High-performance anion exchange chromatography (HPAEC) with pulsed amperometric detection (PAD) was performed with a Dionex ICS-6000 HPIC system (Thermo Fisher Scientific) for quantification of α-1,4-glucans and acarbose. Carbohydrates were separated at 35 °C with a Dionex CarboPac PA100 column (250 × 4 mm, 8.5 µm, Thermo Fisher Scientific) coupled with a Dionex CarboPac PA100 Guard column (250 × 4 mm, 8.5 µm, Thermo Fisher Scientific). Pulsed amperometric detection was performed with the gold, carbo, quad waveform using a non-disposable gold electrode and an AgCl reference electrode at a system temperature of 30 °C. Flow rate was set at 1 mL min^−1^. An amount of 20 µL of the diluted sample (40- and 400-fold) was injected to the system. Eluent A (166 mM ammonium hydroxide) and eluent B (1 M sodium acetate with 166 mM ammonium hydroxide) were applied using the following gradient: 6.5 min 10% B, 31.5 min 25% B, 34.0 min 25% B, 44.0 min 10% B. Evaluation of the data was executed with Chromeleon Chromatography Data System 7.2.10 software (Thermo Fisher Scientific).

### 2.9. LC-ESI-MS Analysis

Mass spectrometry (MS) was used for the specific identification of acarviosyl metabolites and was performed with a micrOTOF-Q hybrid quadrupole/time-of-flight (QTOF) mass spectrometer (Bruker Daltonics, Billerica, MA, USA) equipped with an electrospray ionization (ESI) source. MS was coupled to an UltiMate 3000 HPLC system (Thermo Fisher Scientific). An amount 2 µL of the sample was injected and separated with a flow rate of 0.2 µL min^−1^ using an Accucore 150-Amide-HILIC column (150 × 2.1 mm, 2.6 µm, Thermo Fisher Scientific). Eluent A was an aqueous ammonium formate (10 mM, pH 4.6) solution and eluent B was acetonitrile. The separation gradient was as follows: 0 min 80% B, 20 min 15% B, 22.5 min 15% B, 25 min 80% B, 40 min 80% B. The MS detection range was set at *m/z* 400–1800 in positive ionization mode. Evaluation of the data was performed with Bruker Compass DataAnalysis 4.2 software (Bruker Daltonics).

## 3. Results

### 3.1. Simultaneous Assembly and Disassembly of Linear α-1,4-Glucans

AcbQ is one of the uncharacterized enzymes of *Actinoplanes* sp. SE50/110 and encoded within the acarbose biosynthesis gene cluster [22,30]. BLASTP analysis has shown similarities to MalQ proteins from other strains but also from its own *Actinoplanes* strain (ACSP50_7587, 42% identity, 56% similarity) [14]. MalQ belongs to the enzyme class of 4-α-glucanotransferases (EC 2.4.1.25), which catalyzes the transfer of an α-1,4-glucan moiety to the reducing end of an acceptor (UniProtKB). To analyze the enzymatic function of AcbQ, the codon-optimized gene was cloned into the pJOE5751.1 vector (Appendix A) and expressed in *E. coli* JM109 (Appendix A). The recombinant protein AcbQ (75 kDa) was purified via His_6_-tag (Figure 2A) and its glucanotransferase activity was tested. To identify the substrate spectrum, AcbQ was incubated with α-1,4-glucans of different lengths (maltose (G2), maltotriose (G3) or maltotetraose (G4)) (Figure 2B). An in vitro assay with heat-inactivated AcbQ and maltotriose was used as negative control. TLC analysis of the assay with maltotriose showed the synthesis of new products with shorter and longer chain lengths than the substrate (Figure 2B, blue boxes). The new products had a similar retention time to glucose (G1) and maltopentaose (G5), indicating a simultaneous assembly and disassembly. In assays with maltose or maltotetraose as substrate, no glucanotransferase activity could be detected by TLC analysis.

### 3.2. Substrate Preference of the 4-α-Glucanotransferase AcbQ

To characterize the substrate preference of AcbQ, the protein was incubated with a broad range of different substrates. The enzymatic activities were analyzed using the hexokinase/glucose-6-phosphate dehydrogenase coupled enzyme assay (Figure 3A) [25]. Glucose accumulation was measured in this coupling assay because glucose is described as a cleavage product from MalQ reactions [24,25]. The reactions of the coupling assay were started by addition of AcbQ. The cleavage of glucose led to an increase in absorbance at 340 nm due to the reduction of NADP^+^ to NADPH. The increase of NADPH is stoichiometrically equivalent to the glucose amount released by AcbQ. The initial velocity was calculated from the increase of glucose concentration during the first ten minutes. The reaction mixture with inactivated AcbQ was used as the negative control.

Different groups of single oligosaccharides were tested (Figure 3). In the first group, oligosaccharides with α-1,4-glycosidic bonds from maltose (G2) to maltohexaose (G6) were used as substrate. Here, only low amounts of glucose were formed for maltose (≈0.3 µM min^−1^) and maltotetraose (≈0.7 µM min^−1^). Maltotriose achieved the highest initial velocity of ≈21.3 µM min^−1^. Furthermore, oligosaccharides with α-1,1-glycosidic bond such as trehalose (Tre) and with α-1,2-glycosidic bond such as sucrose (Suc) showed no glucose release during the incubation with AcbQ. Other tested single substrates were acarbose (Acb) and acarbose 7-phosphate (Acb-7P, prepared by AcbK enzyme reaction, Appendix A), which also exhibited no effect. In comparison to the enzyme reactions in which only maltotriose was added, combinations with acarbose or acarbose 7-phosphate delivered initial velocities decelerated by 38% (Acb + G3) and 45% (Acb-7P + G3). Although maltose, maltotetraose or acarbose as single substrates resulted in little or no release of glucose, this could be increased by combining them (Acb + G2: 10-fold, Acb + G4: 4-fold). In general, combinations with acarbose 7-phosphate resulted in no increase of the initial velocity (Figure 3B).

### 3.3. Combination of α-1,4-Glucans Results in Different Elongated Glucan Chains

In a next step, the product formation of AcbQ with α-1,4-glucans as substrates was analyzed. Therefore, the α-1,4-glucans (G2, G3, G4), which led to a glucose formation in the hexokinase/glucose-6-phosphate dehydrogenase coupled enzyme assay (Figure 3B), were incubated with AcbQ. The enzyme reactions were analyzed by HPAEC-PAD after five hours of incubation. The concentration of glucans with chain lengths from one to seven glucose molecules (G1–G7) was measured and calculated using standard curves. The results are displayed in Figure 4, including all substrates and products for this chain length range.

Incubation of AcbQ with maltose showed a small amount of glucose, as shown previously in Figure 3B. No other newly formed products were detected. In contrast to maltose, incubation of AcbQ with maltotriose formed new products with two and four more glucose molecules (G5 and G7) (Figure 4). Almost the same amount of glucose as the newly formed products maltopentaose (G5) and maltoheptaose (G7) together was produced. Moreover, a small amount of maltose and maltotetraose was measured. Maltotetraose as a substrate leads to small amounts of glucose and a slightly larger amount of maltose. The main product with a prolonged chain length was maltohexaose. In addition, maltotriose as the preferred substrate of AcbQ (Figure 3B) was combined with maltose, maltotetraose and maltopentaose, respectively (Figure 4). The combination of these glucans resulted in a higher accumulation of long chain glucans like maltohexaose (assay with G3 and G4) and maltoheptaose (assay with G3 and G5).

### 3.4. Detection of New Peaks in Assays with Acarbose and α-1,4-Glucans

In addition to Section 3.3, product formation of assays containing acarbose with or without a single α-1,4-glucan (G2, G3 or G4) catalyzed by AcbQ was evaluated using HPAEC-PAD analysis (Appendix A). The distribution of glucans with different chain lengths (G1–G5) was as previously shown in Figure 4. In comparison to maltotriose as a single substrate, less maltoheptaose has been synthesized, while the final maltotriose concentration was the same in enzyme reactions with acarbose and maltotriose. Moreover, glucose release was increased by a factor of 1.6 in this reaction (Appendix A). This most likely implies that maltotriose was also used to form another product. The chromatogram of the assay with two substrates confirmed this assumption, as new peaks appeared (Figure 5). For better identification of the new peaks (red boxes), the respective enzyme assay was measured in two different dilutions (40- and 400-fold) using HPAEC-PAD analysis. One new peak (≈8.6 min) occurred shortly after the maltopentaose (G5) peak, and the second peak (≈15.8 min) occurred after the maltoheptaose (G7) peak. Since the area of peak 1 is larger than of the G5 peak (Figure 5B), this suggests that more of the new product was synthesized than maltopentaose. The exact concentration could not be calculated because there was no standard available. Similar new peaks have also been detected in chromatograms of assays with acarbose and maltose or maltotetraose (chromatograms not shown).

### 3.5. Detection of Elongated Acarviosyl Metabolites by LC-ESI-MS

To confirm the newly detected products of AcbQ assays with acarbose and α-1,4-glucans (G2, G3 and G4), LC-ESI-MS analyses of these enzyme reactions were performed. The assumption that acarbose serves as an acceptor for glucanotransferase activities and is linked to activated α-1,4-glucans resulting in elongated acarviosyl metabolites like acarviosyl-maltotriose (Ac-G3) and acarviosyl-maltotetraose (Ac-G4) shall be elucidated. Longer acarviosyl metabolites are already described as by-products of acarbose production in *Actinoplanes* sp. SE50/110 [14,17,20]. An overview of potential acarviosyl metabolites and their masses is given in Table 2.

The detected unknown peaks in the AcbQ assay with acarbose and maltotriose (Figure 5B) could be identified as acarviosyl-maltotetraose (Ac-G4, *m/z* 970.36 [M + H]^+^) and acarviosyl-maltohexaose (Ac-G6, *m/z* 1294.47 [M + H]^+^) by LC-ESI-MS (Figure 6). These results show that mainly two or four glucose molecules were attached to acarbose. Furthermore, a small peak of acarviosyl-maltotriose (Ac-G3, *m/z* 808.317 [M + H]^+^) could be detected in the corresponding extracted-ion chromatogram (EIC) (chromatogram not shown, Table 3). In enzyme reactions with acarbose and maltose, acarviosyl-maltotriose was the main product, followed by acarviosyl-maltotetraose (Appendix A). When acarbose was combined with maltotetraose, acarviosyl-maltotetraose and acarviosyl-maltopentaose could be identified as main products (Appendix A, Table 3). The newly formed acarviosyl metabolites demonstrate that acarbose is extended at its reducing end due to the sugar residue. To make sure that the transferred glucan chains do not bind at the non-reducing end of the C_7_-cyclitol ring, pre-tests were performed using valienol as an acceptor. No extended valienol with one or more glucose molecules could be detected by LC-ESI-MS, confirming that a sugar residue is necessary for AcbQ transfer reactions.

### 3.6. Analysis of Modification of Phosphorylated Acarbose by AcbQ

Afterwards, AcbQ in vitro assays have been conducted with acarbose 7-phosphate in combination with maltose, maltotriose or maltotetraose to analyze if phosphorylated acarbose can serve as an acceptor. This assumption could be confirmed, since acarviosyl-maltotetraose 7-phosphate (Ac-G4-7P, *m/z* 1050.33 [M + H]^+^) was found in AcbQ assay with acarbose 7-phosphate and maltotriose (Figure 7, Table 3). In this case, two glucose molecules have been added to acarbose 7-phosphate. In addition, acarviosyl-maltotetraose 7-phosphate was also observed in enzyme reactions with acarbose 7-phosphate and maltose (Appendix A), or with maltotetraose (Appendix A). Longer phosphorylated acarviosyl metabolites could not be found. All identified acarviosyl metabolites in AcbQ assays with acarbose or acarbose 7-phosphate and α-1,4-glucans are summarized in Table 3.

## 4. Discussion

Acarbose production from *Actinoplanes* plays an important role industrially and pharmaceutically [14,31]. As part of the production of high-purity pharmaceuticals, the elimination of minor compounds is an important and cost-intensive step [32,33]. Therefore, the study of the formation of these minor components is of high interest, making the putative acarbose 4-α-glucanotransferase AcbQ of particular interest [14,17]. In this study, the specific function of AcbQ regarding acarbose modification is shown and confirmed by in vitro assays. In addition to the structural similarity to MalQ, a high functional similarity was determined. MalQ cleaves mainly glucose from short glucans and forms a transition state with the activated glucanosyl moiety, which is subsequently attached to a usually longer glucan acceptor [25,34]. Furthermore, MalQ uses glucans with chain lengths of at least two glucose molecules whereby maltotetraose is the preferred substrate of, e.g., MalQ from *E. coli* [35]. All lengths of α-1,4-glucans and additionally glucose can be used as acceptors [34,35]. This general mechanism of MalQ concerning donor, acceptor and released glucanosyl moiety varies in different publications over time and is not completely solved yet [23,24,26].

First enzyme tests analyzed by TLC showed a coupled assembly and disassembly of α-1,4-glucans catalyzed by AcbQ due to the accumulation of glucose and maltopentaose in assays with maltotriose (Figure 2B). In a next step, substrate preferences were analyzed based on the glucose release in which the assay with maltotriose showed the highest initial velocity (Figure 3). The maltosyl moiety was connected to AcbQ in a transition state, and then might be linked to the acceptor maltotriose (Figure 4) [34]. Afterwards, newly formed maltopentaose served as an additional acceptor, resulting in maltoheptaose as a new product. Maltotetraose was a non-preferred donor due to remaining high amounts of the initial substrate and only a small quantity of newly formed products (Figure 4). However, maltotetraose could serve as an acceptor in combination with maltotriose, and maltohexaose could be produced in addition to maltopentaose (Figure 4). Furthermore, more maltose than glucose was cleaved by AcbQ for activation when maltotetraose was the substrate. This is different from MalQ [34] and could be a side activity to provide the cell not only with glucose but also with maltose. Since maltose is also needed for acarbose formation catalyzed by AcbI [10], maltose turned out to be a non-preferred substrate (Figure 4). In addition, substrates with either α-1,1-glycosidic (trehalose) or α-1,2-glycosidic (sucrose) bonds could not be utilized by AcbQ (Figure 3), as described previously for MalQ [34]. Accordingly, enzyme reactions of 4-α-glucanotransferases such as AcbQ are binding specific for α-1,4-glycosidic bonds as they also occur in the pseudo-tetrasaccharide acarbose (Figure 1).

In further experiments, acarbose was incubated with AcbQ to specify its activity and its influence on acarbose production. It was shown that acarbose does not serve as a single substrate, as no reaction could be detected by mass spectrometry. This contrasts with the combination of acarbose with α-1,4-glucans (G2, G3 or G4) as a substrate mix. Here, a clear formation of long-chain acarviosyl metabolites could be demonstrated (Figure 6). Acarbose appears to serve as an acceptor for the transfer of glucanosyl units. For example, acarviosyl-maltotetraose and acarviosyl-maltohexaose were formed from acarbose and maltotriose since the maltosyl unit was transferred to acarbose or to already newly formed products (Figure 6). When maltose or maltotetraose was incubated with acarbose, the product spectrum could be extended to include acarviosyl-maltotriose and acarviosyl-maltopentaose, respectively (Table 3, Appendix A). These are all elongated acarviosyl metabolites derived from acarbose, which is formed by the pseudo-glycosyltransferase AcbS in the final step of acarbose biosynthesis [10]. Afterwards, AcbQ is involved in the modification of acarbose in addition to the acarbose 7-phosphotransferase AcbK within the cell [13,21].

AcbK is an important intracellular enzyme with a protective function because it catalyzes the phosphorylation of acarbose [13]. *Actinoplanes* sp. SE50/110 produces acarbose to inhibit α-amylases and α-glucosidase from other organisms. However, its own amylases are also essential for its maltose and maltodextrin pathway and should not be affected, which is enabled with phosphorylated acarbose [13]. Thus, it can be assumed that mainly acarbose 7-phosphate is present in the cell and was therefore also tested as an acceptor of AcbQ (intracellular protein) [21,27,36]. Interestingly, this assumption could be confirmed, and the formation of extended phosphorylated acarviosyl metabolites was detected (Figure 7). Mainly acarviosyl-maltotetraose 7-phosphate was identified (Table 3); other products were found only in small quantities. One reason for the low product identification could be a slower reaction rate than with acarbose or with glucans only. Nevertheless, it should be considered that the enzyme reactions all occurred in vitro and thus only partially reflect the biological, intracellular environment of *Actinoplanes* sp. SE50/110. Most likely, acarbose 7-phosphate is the main acceptor since the reaction takes place in the cell (membrane-associated) and acarbose is less relevant in vivo [21]. For example, steric hindrance from the additional phosphorylation compared to acarbose can also lead to a more specific but also slower enzyme reaction [37,38].

AcbQ is part of the postulated carbophore model in *Actinoplanes* sp. SE50/110, which describes an additional sugar transfer using acarbose as a transport vehicle [14]. In this model, acarbose is loaded extracellularly with sugar molecules. The elongated acarviosyl metabolites are reimported and released from acarbose [27,39]. Regarding this model, AcbQ disassembles long acarviosyl metabolites by converting the long α-1,4-glucan moiety into shorter glucans [14]. In this study, the presumed role of AcbQ can be extended, as it was shown that longer acarviosyl metabolites are formed by AcbQ. In addition, *Actinoplanes* sp. SE50/110 features the glycosyltransferase AcbD as an extracellular part of the carbophore model [27,39]. The elongation of acarbose catalyzed by AcbD has already been proven [20]. In contrast to *Actinoplanes* sp. SE50/110, the acarbose biosynthesis gene cluster (*gac* cluster) of *Streptomyces glaucescens* GLA.O does not include an AcbD homolog, but a putative acarbose α-1,4-glucanotransferase GacQ (AcbQ homolog, SGLAU_01030, 59% identity, 67% similarity) [19,40]. This supports the idea that AcbQ can elongate and shorten acarviosyl metabolites depending on the chain lengths of the transferred glucan, which is capable of replacing the function of AcbD.

## 5. Conclusions

This study on the acarbose 4-α-glucanotransferase AcbQ of *Actinoplanes* sp. SE50/110 carried out the glucanotransferase reaction with maltotriose as the preferred substrate and acarbose as the specific acceptor in the acarbose biosynthesis pathway. However, in vivo acarbose 7-phosphate is supposed to be the more relevant acceptor of AcbQ, which is shown in this study. Longer acarviosyl metabolites (α-acarviosyl-(1,4)-maltooligosaccharide) with up to four additional glucose molecules were identified after the glucanotransferase reaction. These longer acarviosyl metabolites are relevant in the industrial context as undesirable by-products of acarbose production, which makes AcbQ an interesting target for further studies on genomic and molecular biological levels.

## Figures and Tables

**Figure 1 microorganisms-11-00848-f001:**
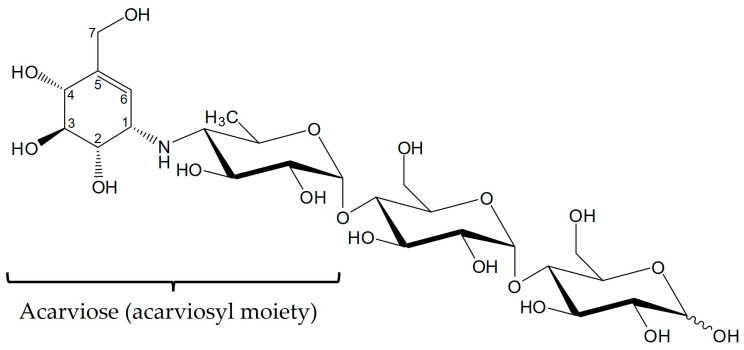
Structural formula of acarbose. Acarbose consists of an acarviosyl moiety and a maltose residue.

**Figure 2 microorganisms-11-00848-f002:**
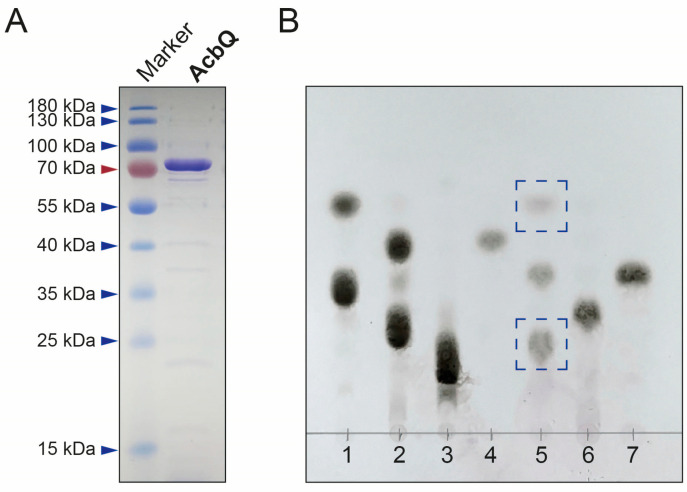
Purification and functional analysis of 4-α-glucanotransferase AcbQ of *Actinoplanes* sp. SE50/110. (**A**) SDS-PAGE analysis of recombinant His-tagged AcbQ protein from *E. coli* JM109. (**B**) TLC analysis of enzymatic function of AcbQ. Lanes 1–7 represent the separation of standard sugars and in vitro assays of AcbQ in combinations with different substrates. 1: Standard sugars glucose (G1) and maltotriose (G3). 2: Standard sugars maltose (G2) and maltotetraose (G4). 3: Standard sugar maltopentaose (G5). 4: Assay with G2. 5: Assay with G3. 6: Assay with G4. 7: Assay with inactivated AcbQ and G3 (negative control). Blue boxes indicate the formation of new products.

**Figure 3 microorganisms-11-00848-f003:**
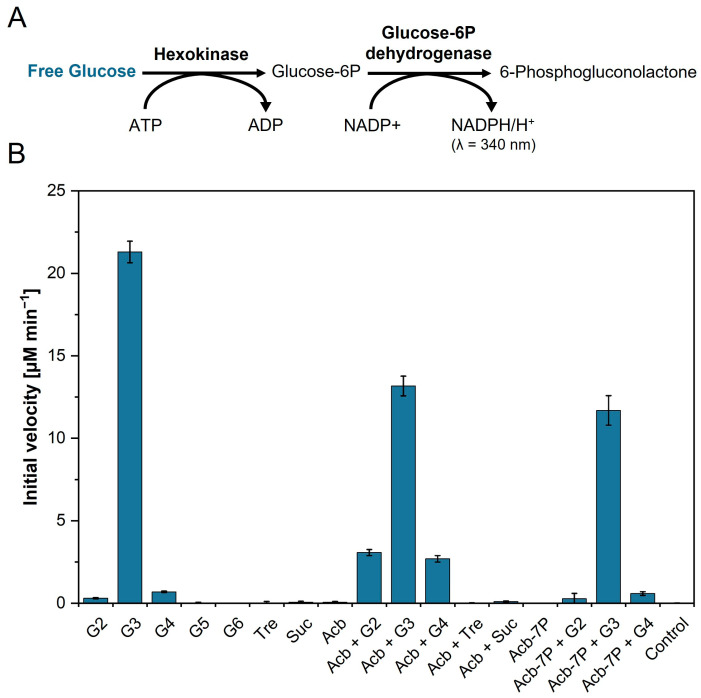
Biochemical characterization of substrate preference of AcbQ. (**A**) Spectrophotometric coupled enzyme reaction with a hexokinase and glucose-6-phosphate dehydrogenase to detect the formation of glucose. (**B**) Initial velocity of AcbQ with different substrates. Control reaction was performed with inactivated AcbQ (*n* = 3).

**Figure 4 microorganisms-11-00848-f004:**
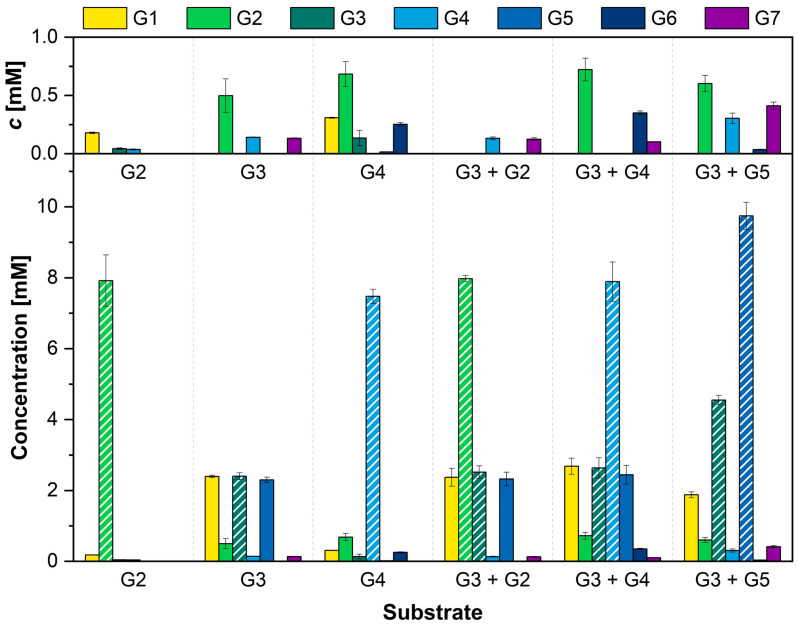
HPAEC-PAD analysis of AcbQ reaction products. Reaction mixtures contained maltose (G2), maltotriose (G3), maltotetraose (G4) or combinations of two α-1,4-glucans: G3 + G2, G3 + G4, G3 + G5. Measurements of the product spectrum (glucans with chain lengths G1–G7) of each assay combination are shown. Glucans that were added as substrates to the assay are marked with stripes. The inset shows a magnification of glucans only with small concentrations below 1 mM (*n* = 3).

**Figure 5 microorganisms-11-00848-f005:**
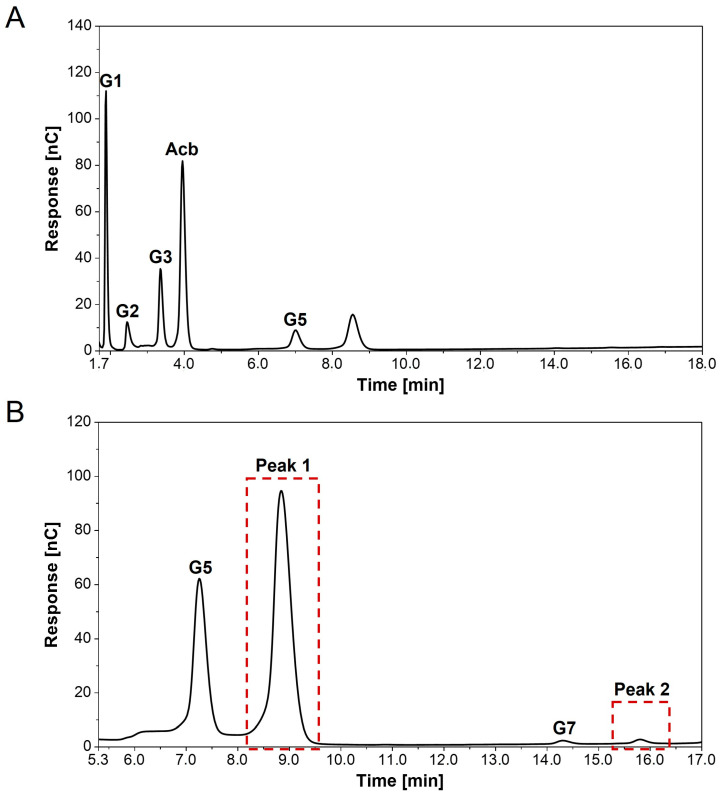
HPAEC-PAD chromatograms of in vitro AcbQ assay with acarbose and maltotriose. Detection of α-1,4-glucans with different chain lengths (G1–G7). (**A**) A 400-fold dilution of enzyme reaction. (**B**) A 40-fold dilution of enzyme reaction. Red boxes mark the two new identified peaks.

**Figure 6 microorganisms-11-00848-f006:**
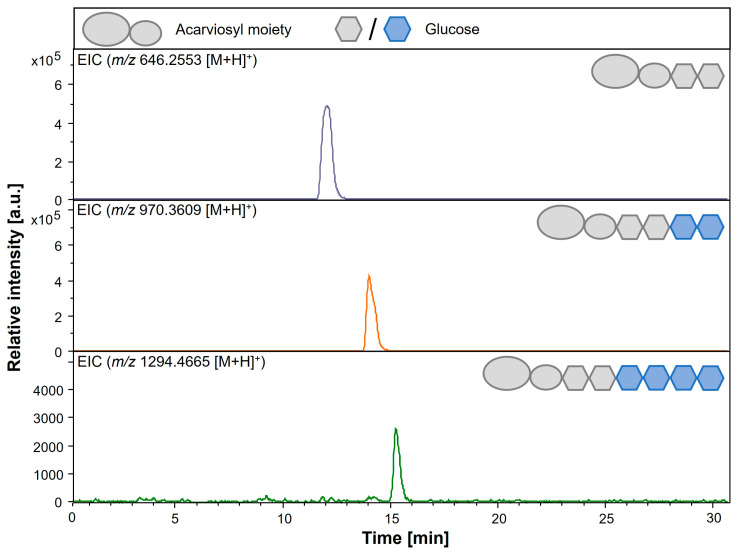
LC-ESI-MS analysis of AcbQ products. Reaction mixture was incubated with acarbose and maltotriose. ESI (+) EIC for acarbose *m/z* 646.2553 (**top** chromatogram), ESI (+) EIC for acarviosyl-maltotetraose *m/z* 970.3609 (**middle** chromatogram), ESI (+) EIC for acarviosyl-maltohexaose *m/z* 1294.4665 (**bottom** chromatogram). Chromatograms contain the schematic illustration of the respective acarviosyl metabolite. Legend to the illustrations is on top.

**Figure 7 microorganisms-11-00848-f007:**
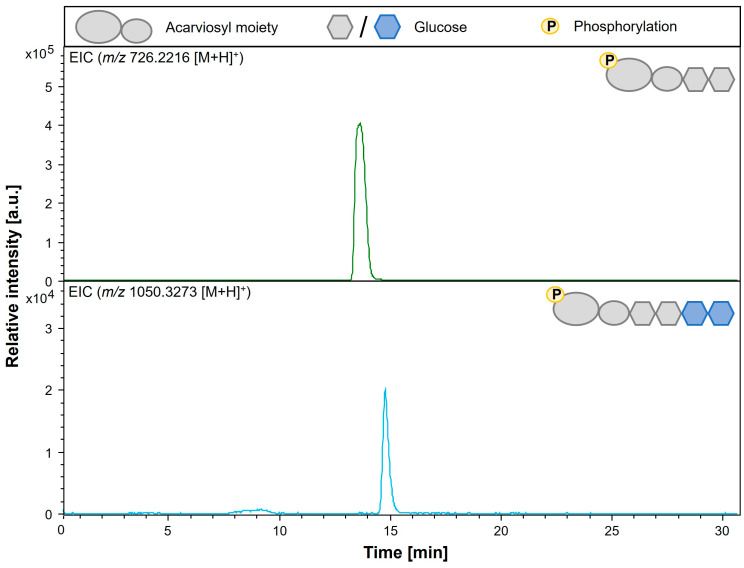
LC-ESI-MS analysis of AcbQ products. Reaction mixture was incubated with acarbose 7-phosphate and maltotriose. ESI (+) EIC for acarbose 7-phosphate *m/z* 726.2216 (**top** chromatogram), ESI (+) EIC for acarviosyl-maltotetraose 7-phosphate *m/z* 1050.3273 (**bottom** chromatogram). Chromatograms contain the schematic illustration of the respective acarviosyl metabolite. Legend to the illustrations is on top.

**Table 1 microorganisms-11-00848-t001:** Structure of acarbose and related acarviosyl metabolites (Ac = acarviose, Glc = glucose, Fru = fructose, Man = mannose). Structures of components 3–4c according to Wehmeier and Piepersberg [14].

Name	Specific Name (Used in This Study)	Structure
Component 1	Acarviose	Ac
Component 2	Acarviosyl-glucose	Ac-1,4-Glc
Component 3	Acarviosyl-maltose (Acarbose)	Ac-1,4-Glc-1,4-Glc
Component 3a		Ac-1,4-Glc-1,4-Fru
Component 3b		Ac-1,4-Glc-1,4-(1-*epi*-valienol)
Component 3c		Ac-1,4-Glc-1,1-Glc
Component 3d		Ac-1,4-Glc-1,4-Man
Component 4a		Ac-1,4-Glc-1,4-Glc-1,4-Fru
Component 4b	Acarviosyl-maltotriose	Ac-1,4-Glc-1,4-Glc-1,4-Glu
Component 4c		Ac-1,4-Glc-1,4-Glc-1,1-Glu
Component 5–7	Acarviosyl-maltotetraose/ -pentaose/-hexaose	Ac-1,4-Glc-1,4-Glc-1,4-Glu-1,4-Glu/ -1,4-Glu/-1,4-Glu

**Table 2 microorganisms-11-00848-t002:** Overview of acarviosyl metabolites derived from acarbose. Mass list of phosphorylated and non-phosphorylated acarviosyl metabolites including their sum formula and [M + H]^+^ mass.

Acarviosyl Metabolites	Abbreviation	Sum Formula	[M + H]^+^ *m/z*
Acarviosyl-maltose (Acarbose)	Acb	C_25_H_43_NO_18_	646.2553
Acarviosyl-maltotriose	Ac-G3	C_31_H_53_NO_23_	808.3081
Acarviosyl-maltotetraose	Ac-G4	C_37_H_63_NO_28_	970.3609
Acarviosyl-maltopentaose	Ac-G5	C_43_H_73_NO_33_	1132.4138
Acarviosyl-maltohexaose	Ac-G6	C_49_H_83_NO_38_	1294.4665
Acarviosyl-maltose 7-phosphate (Acarbose 7-phosphate)	Acb-7P	C_25_H_44_NO_21_P	726.2216
Acarviosyl-maltotriose 7-phosphate	Ac-G3-7P	C_31_H_54_NO_26_P	888.2744
Acarviosyl-maltotetraose 7-phosphate	Ac-G4-7P	C_37_H_64_NO_31_P	1050.3273
Acarviosyl-maltopentaose 7-phosphate	Ac-G5-7P	C_43_H_74_NO_36_P	1212.3801
Acarviosyl-maltohexaose 7-phosphate	Ac-G6-7P	C_49_H_84_NO_41_P	1374.4329

**Table 3 microorganisms-11-00848-t003:** Overview of detected acarviosyl metabolites in AcbQ enzyme assays by LC-ESI-MS. Acarbose and acarbose 7-phosphate were combined with different α-1,4-glucans (G2, G3 or G4). Identified acarviosyl metabolites are marked with +.

	**Ac-G3**	**Ac-G4**	**Ac-G5**	**Ac-G6**
Acb + G2	+	+		
Acb + G3	(+) ^1^	+		+
Acb + G4	(+) ^1^	+	+	
	**Ac-G3-7P**	**Ac-G4-7P**	**Ac-G5-7P**	**Ac-G6-7P**
Acb-7P + G2	(+) ^1^	+		
Acb-7P + G3		+		
Acb-7P + G4		+		

^1^ Acarviosyl metabolites that could be detected in a low amount.

## Data Availability

Not applicable.

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
