# Peer review of "The 4-α-Glucanotransferase AcbQ Is Involved in Acarbose Modification in Actinoplanes sp. SE50/110"

_microorganisms, 2023, doi:10.3390/microorganisms11040848_

Round 1

Author Response

Please find a document detailing on all comments made by the reviewers 1 and 3 attached.

Reviewer 2 Report

The manuscript can be accepted 

Author Response

Thank you for your nice review!

Reviewer 3 Report

The manuscript "The 4-α-Glucanotransferase AcbQ is Involved in Acarbose Modification in Actinoplanes sp. SE50/110" is devoted to the biosynthesis of acarbose. Despite the significant industrial role of this biosynthetic gene cluster, the exact functions of some encoded proteins, e.g. AcbQ, remain unclear. AcbQ was shown to be a 4-α-glucanotransferase, involved in elongation and formation of by-products during acarbose synthesis. The study was performed on recombinant proteins AcbQ and AcbK, various glucans were tested as substrates for AcbQ in vitro. The manuscript is suitable for publication in Microorganisms.

Minor comments:

1. Figure 4 is not clear. The colors of the small bars are very similar and the results on the main figure are therefore unreadable. It is unclear that the upper zoomed-in section represents the smallest bars of the main section, with the large bars removed.

2. In Table 2, there is no need for both neutral mass and [M+H], one of these columns should be removed.

Author Response

(The authors gave the same response as above.)
